# Evaluating methods for reconstructing large gaps in historic snow depth time series

Johannes Aschauer[1] and Christoph Marty[1]

[1]WSL Institute for Snow and Avalanche Research SLF

**Correspondence:** Johannes Aschauer (johannes.aschauer@slf.ch)

**Abstract.** Historic measurements are often temporally incomplete and may contain longer periods of missing data whereas climatological analyses require continuous measurement records. This is also valid for historic manual snow depth (HS) measurement time series, where even whole winters can be missing in a station record and suitable methods have to be found to reconstruct the missing data. Daily in-situ HS data from 126 nivo-meteorological stations in Switzerland in an altitudinal range of 230 to 2536 m above sea level is used to compare six different methods for reconstructing long gaps in manual HS time series by performing a "leave-one-winter-out" cross-validation in 21 winters at 33 evaluation stations. Synthetic gaps of one winter length are filled with bias corrected data from the best correlated neighboring station (BSC), inverse distance weighted (IDW) spatial interpolation, a weighted normal ratio (WNR) method, Elastic Net (ENET) regression, Random Forest (RF) regression and a temperature index snow model (SM). Methods that use neighboring station data are tested in two station networks with different density. The ENET, RF, SM and WNR methods are able to reconstruct missing data with a coefficient of determination ($r^2$) above 0.8 regardless of the two station networks used. Median RMSE in the filled winters is below 5 cm for all methods. The two annual climate indicators, average snow depth in a winter (HSavg) and maximum snow depth in a winter (HSmax), can be well reproduced by ENET, RF, SM and WNR with $r^2$ above 0.85 in both station networks. For the inter-station approaches, scores for the number of snow days with HS≥1 cm (dHS1) are clearly weaker and except for BCS positively biased with RMSE of 18-33 days. SM reveals the best performance with $r^2$ of 0.93 and RMSE of 15 days for dHS1. Snow depth seems to be a relatively good-natured parameter when it comes to gap filling of HS data with neighboring stations in a climatological use case. However, when station networks get sparse and if the focus is set on dHS1, temperature index snow models can serve as a suitable alternative to classic inter-station gap filling approaches.

## 1 Introduction

Climatological analyses require continuous measurement series of meteorological data. Unluckily, historical measurement series are prone to contain periods of missing data. Longer data gaps can for example originate from temporally abandoning a measurement site, not properly reported measurements or archiving errors. Therefore, periods of missing data ideally need to be interpolated prior to execution of any analysis. This is also valid for manual snow depth (HS) measurement time series. For example, many instances of a whole winter of missing data are present in the manual station HS data records in Switzerland. On the other hand, long-term continuous records of HS are for example necessary to perform climatological trend analyses

(e.g. Matiu et al., 2021), to verify modeling studies (e.g. Olefs et al., 2020) or to calculate return levels of extreme events for constructional guidelines (e.g. Marty and Blanchet, 2012).

A number of studies have evaluated and compared methods for reconstructing missing data mostly for the two variables temperature and precipitation (e.g. Kanda et al., 2018; Woldesenbet et al., 2017; Yozgatligil et al., 2013; Kemp et al., 1983). For longer gaps, usually inter-station approaches are used where missing data of one station is imputed with the help of one or more neighboring stations (Massetti, 2014). For this purpose, most often multiple regressions, weighted averages or ratios of average values between the neighboring station and the station to be filled are used (Woldesenbet et al., 2017; Tardivo and Berti, 2012; Auer et al., 2007). More recently, also machine learning approaches have been used to estimate missing values (Kim and Pachepsky, 2010; Kashani and Dinpashoh, 2012).

Snow depth is the result of an interplay between temperature and precipitation as well as the radiation driven energy budget. Therefore, it is unclear if the methods developed for the reconstruction of other meteorological parameters are also easily applicable for snow depth time series. Additionally, for inter-station approaches there might be the problem of different relationships during accumulation and ablation phase between stations which could hinder such approaches (Bales et al., 2018). This might be especially true for stations at different elevations. Inter-station approaches are limited by the fact that a suitable set of reference stations needs to be available. Additionally, different predominant macro-scale weather patterns from one winter to the other can lead to the violation of the assumption that relationships between stations are stationary. If other meteorological parameters have been continuously measured in the period of missing HS at the target station, HS can also be derived from these parameters with snow models. For the climatological use case where measured data is often limited by the number of input variables and the temporal resolution, temperature-index models can be used for this task as they only need daily precipitation and mean temperature as input variables. Although temperature-index models are very simplistic and for example neglect effects such as snow redistribution by wind, they have been used in snow climatological impact studies (e.g. Marke et al., 2018; Notaro et al., 2011). Flat field locations which are often characteristic for snow measurement sites are thought to be less affected by such kinds of effects.

Reconstruction of HS data has been done by several studies (e.g. Brown, 1996; Brown et al., 2003; Witmer, 1984; Falarz, 2002; Avanzi et al., 2020). Some of the studies focus on shorter gaps in hourly automatic measured snow data (Avanzi et al., 2020) while other studies focus on monthly means and employ very simple statistical models based on temperature only (Hughes and Robinson, 1993; Brown et al., 1995). For daily data, weighted averages of HS data from neighboring stations are employed (Matiu et al., 2021). Schöner and Koch (2016) use spatial averages and a temperature-index model to reconstruct missing daily HS data in a project of the Austrian meteorological service. However, except for Witmer (1984) who compare spatial interpolation methods for short gaps, no general comparison of different methods for reconstructing long gaps in daily HS time series exists to our knowledge. It remains unclear which methods are most appropriate for climatological analyses because the existing methods from different studies are not easily comparable and also only applicable for specific setups. For climatological analyses covering snow, most often annual or seasonal snow climate indicators are used to evaluate trends and changes in the snow cover rather than the daily values (e.g. Marty, 2008; Beniston, 2012; Buchmann et al., 2021; Marke et al., 2018; Olefs et al., 2020). These snow climate indicators are derived from daily data such as for example mean snow depth or

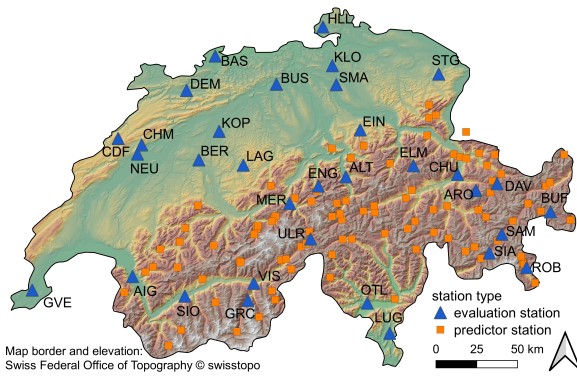

**Figure 1.** Location of evaluation stations (blue triangles) and predictor stations (orange squares) for the cross-validation study. The background color resembles elevation.

duration of the snow cover. However, none of such studies evaluate the influence of missing data and gap filling procedures on these snow climate indicators.

With this study, we perform a quantitative comparison of different methods for reconstructing typical year-long gaps in manual daily HS time series with focus on climatological analyses and the ability to reproduce important annual snow climate indicators. A specific aim is to test the performance of simple temperature-index models, because some gaps occur at the beginning of the measurements series (i.e. in the fist half of the 20th century), when typically no suitable neighboring stations were available. We compare different spatial interpolation methods as well as a simple snow model by imputing synthetic gaps in a "leave-one-winter-out" cross-validation study. The remainder of the paper is structured as follows: Used data and methods are described in Section 2, results are presented and discussed in Section 3 and concluding remarks are given in Section 4.

## 2 Data and methods

We use daily manual snow depth, mean temperature and sum of precipitation data from 126 nivo-meteorological stations in Switzerland. The majority (93) of the stations are primarily measuring snow related variables and not necessarily temperature and precipitation. The stations are either operated by the Swiss Federal Office of Meteorology and Climatology (MeteoSwiss) or by the WSL Institute for Snow and Avalanche Research SLF (SLF) and data is provided by these two institutions. The data covers 21 hydrological years in the period between 1999 until 2020. A hydrological year is defined as the period from September until end of August. The snow depth is measured manually between 7 and 8 a.m. local time each morning from a fixed snow stake an has the date stamp of the day of measurement. Although many stations have already measured snow before 1999, we decide to use only the last 21 years in order to have as many complete and thoroughly quality controlled time series in our station set. The 21 year time period was chosen because we wanted to have a long enough data set on the on hand (containing a few well known snow abundant and snow scarce years) and a common (realistic) length of available

snow depth time series for the training period (see below) on the other hand. The daily sum of precipitation data is covering the period 7 a.m. of the previous day until 7 a.m. local time and has the date stamp of the previous day. Mean temperature is aggregated over the whole day and has no date shift. The change of a HS measurement of date $i$ relative to the preceding measurement is therefore influenced by the precipitation of date $i - 1$ and a combination of the temperature at the two dates

$i$ and $i - 1$. For being able to test methods for reconstructing missing data in a controlled environment, a "leave-one-winter-out" cross-validation is performed. Data for one winter (Nov-Apr) is deleted (gap period) and in case a parameter training is required for the respective method, this is done on the winter data of the remaining 20 winters (training period). Locations of the stations used in the cross validation study can be seen in Fig. 1. We test the spatial interpolation methods in two different station networks in order to assess sensitivity against sparser station networks. Sparser networks can be expected in areas of

the world which are not as densely populated as Switzerland or in earlier times such as e.g. in the mid 20th century when much fewer stations measured snow depth in Switzerland. The dense network contains 33 evaluation stations (blue triangles in Fig. 1) as well as additional 93 neighboring predictor stations (orange squares in Fig. 1) and covers stations in an altitudinal range of 230 to 2536 m above sea level. The sparser network consists of the evaluation stations only and covers an altitudinal range of 273 to 1970 m above sea level. If two stations were situated closer than 3 km to each other, one of the two stations

was excluded from the station sets. In order to test every method at the same set of stations, evaluation stations are chosen thus they have a continuous record for all three variables HS, temperature and precipitation. Therefore, gaps are only filled at the evaluation stations of both station networks. For the stations ARO, DAV and ULR we combined temperature and precipitation data measured by MeteoSwiss with HS data that was measured by the SLF at a close by partner station. Gaps shorter than three days in the HS time series (only rarely occurring) have been filled by linear interpolation. If any variable had longer data gaps

than three days, the corresponding station was excluded from the station data set.

## 2.1    Interpolation methods

### 2.1.1    Selection of neighboring stations for spatial interpolation methods

Six different methods are employed to interpolate a missing winter of snow depth data at a certain station with help of neighboring stations or by using measured meteorological data at the gap station. In case neighboring stations are used as predictors for

reconstructing the missing data, these stations have to be within a radius of 200 km and show an absolute elevation difference of less than 500 m. We choose these limits based on a correlation analysis of Matiu et al. (2021). For all methods which use HS data from neighboring stations, the best $n$ correlated neighboring stations are chosen as predictor stations. If less than $n$ stations meet the constraints defined above, the number of predictor stations is reduced accordingly. To select the best reference stations, Pearson correlations between target station and neighboring stations are computed in the training period only (see

Sec. 2 for definition). The maximum number of potential predictor stations for each of the spatial interpolation methods has been determined in another cross validation study where we varied the number of maximum potential predictor stations from 3 to 25 stations. This sensitivity study is performed only on the complete station network as for the sparse network the maximum

number of 25 stations would not be reached in many example cases. Results of this sensitivity study and the maximum number of potential predictor stations is discussed further in Sec. 3.1.

### 2.1.2 Best correlated station (BCS)

The simplest approach we test for imputing missing data is to directly use HS data from the best correlated neighboring station (BCS). Correlation is calculated in the training period and the constraints defined in Section 2.1.1 have to be fulfilled. As a simple bias correction measure, the data from the BCS is multiplied with the ratio of the mean at the target site to the mean at the BCS calculated in the training period.

### 2.1.3 Inverse distance weighting (IDW)

The inverse distance weighting (IDW) method uses a weighted spatial average of neighboring stations to impute missing values at the target station, neglecting any elevation gradients. Weights are the inverse squared distance of the respective neighboring station to the target station such that

$$\hat{y} = \frac{\sum_{i=1}^{n} \frac{y_i}{d_i^2}}{\sum_{i=1}^{n} \frac{1}{d_i^2}} \tag{1}$$

where $\hat{y}$ is the estimated snow depth at the target station, $n$ is the number of neighboring reference stations, $y_i$ is the snow depth at neighboring station $i$ and $d_i$ is the distance of the neighboring station $i$ to the target station. Imputed values are rounded to the nearest integer. IDW is besides nearest neighbor and non-weighted local averages one of the most often used methods for reconstructing climatological data (Beguería et al., 2019; Kanda et al., 2018).

### 2.1.4 Weighted normal ratio (WNR)

Matiu et al. (2021) use a variation of the weighted normal ratio (WNR) method for filling short and longer gaps (up to few years) in daily snow depth time series. The normal ratio method was first introduced by Paulhus and Kohler (1952) and assumes a constant ratio of the average state of two neighboring stations (Young, 1992; Yozgatligil et al., 2013). Missing values are filled by

$$\hat{y} = \frac{\sum_{i=1}^{n} w_i y_i \frac{\bar{y}}{\bar{y_i}}}{\sum_{i=1}^{n} w_i} \tag{2}$$

where $n$ is the number of neighboring reference stations, $y_i$ is the snow depth at neighboring station $i$, $\bar{y}$ and $\bar{y_i}$ are the mean snow depth at the target station and reference station $i$ in the training period respectively and $w_i$ is the weight of station $i$ based on the vertical distance $Z - Z_i$ calculated as $w_i = e^{-\ln 2(Z-Z_i)^2/250^2}$ which is a Gaussian weight function with a full width at half maximum of 500 m. Reconstructed values are rounded to the nearest cm integer. In order to have equal conditions within our method comparison, the selected neighboring stations do not need to have a correlation coefficient larger than 0.7 with the target contrary to the WNR method used in Matiu et al. (2021).

### 2.1.5 Elastic Net (ENET) regression

As fourth method for reconstructing missing HS data at a target station, we use a multiple linear regression of the HS data from the best correlated neighboring stations. As the neighboring stations often are as well correlated with each other, we use Elastic Net (ENET) regularization to reduce the variance of the model (Zou and Hastie, 2005; Friedman et al., 2010). Elastic Net combines the $l1$ regularization term employed in LASSO (Tibshirani, 1996) and the $l2$ regularization term used in ridge regression (Hoerl and Kennard, 1970) and is thus able to deal with multicollinearity in the predictors. The ratio between $l1$ and $l2$ regularization and the hyperparameter $\alpha$ are optimized in a 5-fold cross validation on the data in the training period. Before fitting and predicting with the model, predictors and target are standard scaled to have a mean of 0 and standard deviation of 1 based on the data in the training period. Reconstructed values are rounded to the nearest cm integer and negative predicted values are clipped to zero.

### 2.1.6 Random Forest (RF) regression

As fifth method we employ Random Forest (RF) regression as a nonlinear combination of neighboring stations. A random forest is an ensemble of decision trees that are drawn from random subsets of the training data (Breiman, 2001). The prediction of the ensemble is the average of the individual trees. We use the best correlated neighboring stations as predictors that satisfy the requirements defined in Section 2.1.1. In order to capture potential different relationships between stations in the course of a snow season, we additionally pass the three seasons early Winter (Nov, Dez), mid Winter (Jan, Feb) and late Winter (Mar, Apr) as a categorical predictor to the model. Prior to fitting the model, this "seasons" predictor is one-hot encoded, whereas the other predictors of neighboring station HS data are standard-scaled as for the elastic net regression (Section 2.1.5). The random forest model has a tree number of 200 and a maximum depth of 70.

### 2.1.7 Snow model (SM)

As last method we make use of a simple snow model (SM). The snow model consists of a temperature-index model which is then coupled to a density model to estimate the snow depth. For estimating snow water equivalent (SWE) in the snowpack, we use the Snow-17 model which uses a temperature-index approach with a seasonally varying melt factor (Anderson, 1973). However, we do not use the density parameterization described in the former reference. Instead, we post-process the SWE time-series of the temperature-index model with a very simple density model. The density model uses an approach based on Martinec and Rango (1991) in which a time dependent density for the different layers in the snowpack is assumed:

$$\rho(t) = \rho_{max} + (\rho_0 - \rho_{max})e^{-t/\tau} \tag{3}$$

Each layer that is identified by an increase in SWE has an initial new snow density $\rho_0$, which is temporally increasing according to Eq. 3 at each time step $t$ until it reaches a maximum density $\rho_{max}$. When SWE decreases during a day, the density model removes layers from top of the snowpack for compensating the loss in SWE. During the cross-validation, only the parameters of the density model $\rho_0$, $\rho_{max}$ and $\tau$ are optimized by grid-searching a predefined reasonable parameter space during the

training period for each station and synthetic gap individually to minimize the root-mean-squared error (RMSE) in the training period. No parameter optimizations are done for the melt and accumulation model and the parameters defined in Anderson (1973) are used. We considered to use a combined temperature of two days to correspond with the interval of precipitation and HS data (see Section 2. However, we found negligible differences in model performance and decided to leave the input data as is to avoid potential smoothing of temperature signals. In contrast to the inter-station methods described above, we apply the snow model over the full hydrological year in order account for snow which has already built up until November. However, scoring is only done in the winter months Nov-Apr.

## 2.2 Evaluation metrics

As score metrics of the reproduced daily HS values we use the RMSE, the coefficient of determination ($r^2$) and the BIAS. The BIAS is calculated as the average error. RMSE and BIAS can be interpreted in the same unit as the HS measurements [cm]. As fourth metric, we use the mean arctangent absolute percentage error (MAAPE) which was introduced by Kim and Kim (2016) as a relative error term (limited to a maximum of 1.6) because of frequent close-to-zero HS values for stations at low elevation which blow up traditional relative error terms such as the mean absolute percentage error. Since we are interested in gap filling for climatological analyses, we additionally test how good the different methods are able to reproduce three snow climate indicators which are frequently used by practitioners. These snow climate indicators are i) the average snow depth in a winter (HSavg) which is widely used to test for trends in snow climatology, ii) the maximum snow depth in a winter (HSmax) which is an important indicator for e.g. prevention measures in engineering, and iii) the number of snow days with HS≥1 cm (dHS1) which has vital importance for ecology and the winter tourism industry.

## 3 Results and Discussion

### 3.1 Number of potential predictor stations

The influence of the maximum number of neighboring stations is displayed in Fig. 2. Boxplots of RMSE and MAAPE scores caluclated in the reconstructed winters are shown for varying numbers of neighboring stations for the different spatial interpolation methods. The methods have been evaluated in the dense station network. IDW shows decreasing performance for both RMSE and MAAPE with increasing number of predictors. The median RMSE increases from 3.9 for one predictor station to 5.6 for 25 predictor stations. For WNR, the median MAAPE is increasing with increasing number of neighboring stations from 0.21 for one neighboring station to 0.37 for a maximum number of 25 neighboring stations. However, WNR performs best in terms of RMSE for a maximum number of 5 neighboring stations with a median RMSE of 3.1. RF and ENET generally show increasing performance with increasing number of predictor stations. For ENET, median RMSE is decreasing from 3.3 for one predictor station to 2.7 for a maximum number of 15 predictor stations. Above 15 predictor stations, a minimal increase of median RMSE to 2.8 is observable. MAAPE scores are decreasing and show a lower spread for increasing maximum number of predictor stations. However, further increase from 15 stations does not yield remarkable differences in median MAAPE

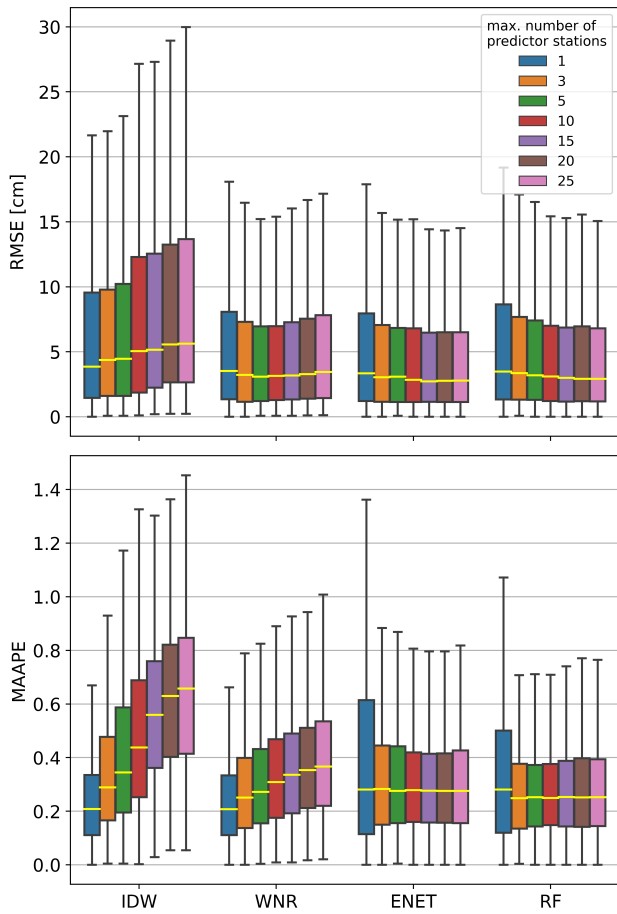

**Figure 2.** Boxplots of RMSE and MAAPE calculated in the individual reconstructed winters with varied maximum number of predictor stations for the spatial interpolation methods. The methods have been applied to the complete station network. For better comparison, outliers are not shown in the boxplots. Note that WNR with one predictor station is equivalent to the BCS method.

and its variance. For RF, RMSE constantly decreases with increasing maximum number of predictor stations from 3.5 for one predictor station to 2.9 for a maximum number of 25 predictor stations. MAAPE scores for RF are insignificantly better for

maximum station numbers of 3, 5 and 10 than for higher maximum station numbers.

Some of the methods are more sensitive to the maximum number of used neighboring stations than others. The deterministic approaches (IDW, WNR) regresses in skill for more stations because more stations introduce unnecessary noise. This is the reason why other studies that use regional averages or simple linear regressions also use only few neighboring stations for reconstructing missing data (e.g. Matiu et al., 2021; Tardivo and Berti, 2014). Regularization measures, which are both

included in the ENET and RF regression, allow to choose the best predictors from a given set of predictor stations. Therefore, overfitting is prevented even for a larger number of predictors with these two methods. Tests on how many predictor stations are

**Table 1.** Selected number of neighboring stations for each method.

| Method | max. # neighboring stations |
|--------|-----------------------------|
| BCS | 1 |
| IDW | 3 |
| WNR | 3 |
| ENET | 15 |
| RF | 10 |
| SM[1] | 0 |

[1] only temperature and precipitation data from the target station is used

influential for the Random Forests showed that only few stations (less than ~5) share the majority of feature importance. The selected number of maximum neighboring stations for the method comparison in Section 3.2.1 and 3.2.2 is mainly based on the median RMSE and MAAPE scores presented earlier. If scores from two different maximum numbers of predictor stations are

approximately equal for one method, we decided to use the lower number of stations to keep the method as simple as possible. Accordingly, we use the maximum numbers of predictor stations listed in Tab. 1 for the comparison of different methods in the following sections.

## 3.2 Method performance

### 3.2.1 Daily values

Predicted daily values are plotted against measured daily values for the different methods and station densities in Fig. 3. Values are aggregated over every filled gap in the cross-validation. The three score metrics $r^2$, RMSE and BIAS are indicated in each panel. For both the sparse and dense station network, ENET regression yields almost always the best results for all score metrics, shortly followed by RF regression and the WNR method. In the dense station network, WNR, ENET and RF have similar score values with RMSE ranging between 6.5 and 7.0, similar $r^2$ of 0.94 and equally small BIAS of 0.06 for ENET

and RF and BIAS of -0.07 for WNR. BCS is shortly following WNR, ENET and RF in the dense station network with $r^2$ of 0.92, RMSE of 7.6 and BIAS of -0.1. IDW is performing poorer than the four aforementioned methods with $r^2$ of 0.85, RMSE of 10.6 and a positive BIAS of 1.78. The Snow Model performs equal to IDW in the dense station network in terms of RMSE and $r^2$ with RMSE of 10.2 and $r^2$ of 0.86. SM predictions are negatively biased with BIAS of -0.74. The SM thus cannot compete with the WNR, BCS, ENET and RF methods in the dense station network. However, the SM (in contrast to IDW) can

compete with the WNR and BCS methods in the sparse station network for which the RMSE increases by ~35% and ~40% compared to the dense station network, respectively. RF and ENET are less sensitive against station network density than the WNR and BCS methods but still performance decreases for decreasing station network density. RMSE in the sparser station network decreases by ~30% compared to the dense station network for RF and ENET. IDW is the most sensitive to station

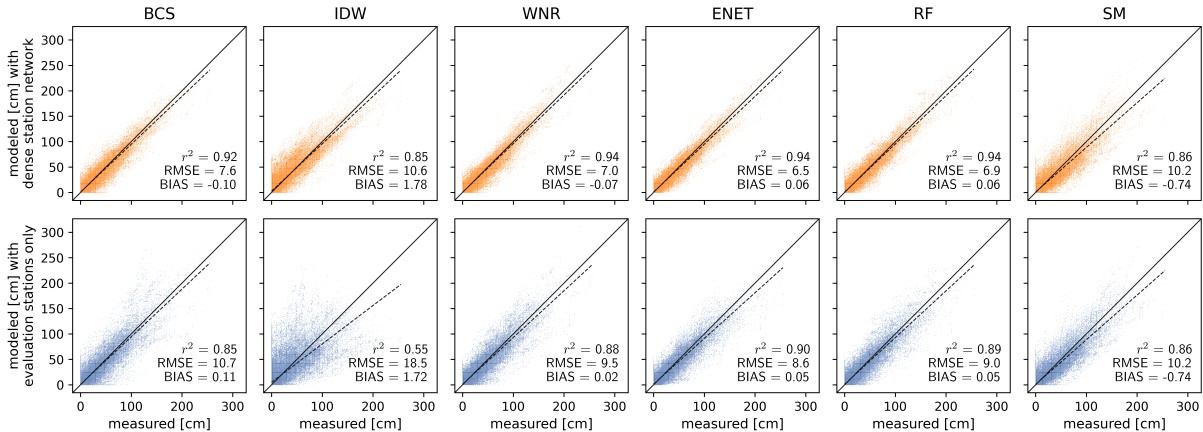

**Figure 3.** Reconstructed daily snow depth values plotted against the measured values for the used methods (columns). Data in the top row is calculated in the full station network, data in the bottom row is calculated using the evaluation stations only. The solid black line represents perfect predictions, the dashed line is a linear fit of predicted versus measured values. The three score metrics coefficient of determination ($r^2$), root-mean-squared error (RMSE), and BIAS are indicated in each panel.

network density. RMSE in the sparse station network increases by ~75% and explained variance is significantly lower with $r^2$
of 0.55 in the sparse station network.

The RMSE scores and BIAS of daily values aggregated over all reconstructed gaps are about double as high as the the median RMSE and BIAS obtained from each gap individually (Fig. A2).

### 3.2.2   Annual snow climate indicators

HSavg, HSmax and dHS1 derived from the reconstructed daily data (Section 3.2.1) are plotted against the same snow climate
indicators derived from the measured data in Fig. 4. The score values BIAS, RMSE and coefficient of determination ($r^2$)
accompanying the data shown in Fig. 4 are listed in Table 2. Absolute errors of the same snow climate indicators derived from reconstructed data versus the HSavg derived from the measured data in the reconstructed winters are shown in Fig. 5.

BCS, WNR, ENET, RF and SM yield unbiased reconstructions of HSavg for both the dense and the sparser station network with BIAS smaller 0.15 cm. For all methods, RMSE for HSavg is about 30 to 40% smaller than the RMSE derived from
the aggregated daily values (see Section 3.2.1) for both the reconstructions from the dense and sparser station network. The absolute error of HSavg and HSmax increases with increasing HSavg for all methods (Fig. 5). However, the increase is much larger for BCS and IDW in the case of the sparser station network.

HSmax derived form the filled gaps shows a ~5-10% lower explained variance than HSavg. RMSE values for HSmax are larger than for HSavg but should be compared cautious because of the different scales of the two snow climate indicators.
BCS, WNR, ENET, RF and the SM yield negatively biased HSmax with biases ranging from -2.3 to -7.4 cm in the dense and -1.6 to -7.4 cm in the sparse station networks, respectively. IDW shows slightly positive BIAS of 2.8 and 2.9 for the dense and

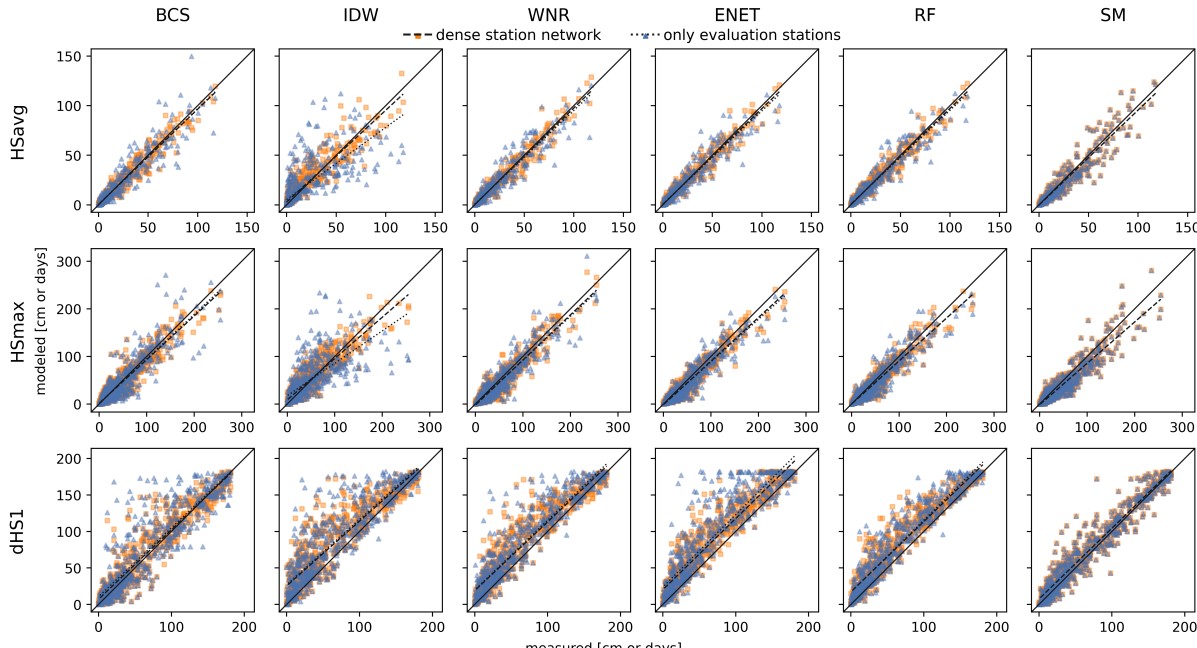

**Figure 4.** Modeled average snow depth in a winter (HSavg, top row), maximum snow depth in a winter (HSmax, middle row) and number of snow days with HS≥1 cm (dHS1, bottom row) of the reconstructed winters from the cross-validation trials versus the respective snow climate indicator value derived from measurements. The columns refer to the different interpolation methods. Orange squares are gaps reconstructed with the complete station network, blue triangles are gaps that have been reconstructed solely using the evaluation stations as depicted in Figure 1. The black line represents perfect predictions. The dashed and dotted lines are linear fits to the data points of the dense and sparse station networks, respectively.

sparse station networks, respectively. Median absolute errors of HSmax are increasing with increasing HSavg for all methods. For BCS and IDW, absolute errors for HSmax are increasingly sensitive to station network density for increasing HSavg. The temporal occurrence of HSmax is consistently well reproduced for all methods in the dense and sparser station network with
median deviations of 0 days, mean deviations ranging from -2.8 to +2.4 days and standard deviations ranging from 31.3 to 37.1 days (see Fig. A3).

The dHS1 is reproduced less precisely than HSavg with ~10-20% lower explained variance $r^2$. All methods apart from BCS and SM strongly overestimate the number of snow days with HS≥1 cm of the reconstructed winters with BIAS from 14.6 to 18.4 days overestimation for the full station network and 16.0 to 23.3 days overestimation for the sparse station network.
However, the BCS also slightly overestimates dHS1 with BIAS of 3.7 and 6.6 days in the dense and sparse station networks, respectively. All methods (except SM by method definition) experience an increase in BIAS of dHS1 in the sparse station network compared to the dense station network. For all methods, the absolute error of dHS1 is largest in winters with HSavg below 40 cm.

**Table 2.** BIAS, RMSE and coefficient of determination ($r^2$) for the three climate metrics HSavg, HSmax and dHS1 reconstructed with the different methods in the dense and sparse station networks as shown in Figure 4.

| | | dense station network | | | | | | evaluation stations only | | | | | |
|---|---|---|---|---|---|---|---|---|---|---|---|---|---|
| | | BCS | IDW | WNR | ENET | RF | SM | BCS | IDW | WNR | ENET | RF | SM |
| **HSavg** | $r^2$ | 0.96 | 0.87 | 0.96 | 0.97 | 0.97 | 0.93 | 0.90 | 0.53 | 0.93 | 0.94 | 0.94 | 0.93 |
| | **RMSE** | 4.45 | 7.92 | 4.15 | 4.04 | 3.91 | 5.98 | 6.80 | 14.87 | 5.81 | 5.37 | 5.32 | 5.98 |
| | **BIAS** | -0.11 | 1.84 | -0.07 | 0.06 | 0.06 | -0.77 | 0.12 | 1.77 | 0.02 | 0.05 | 0.05 | -0.77 |
| **HSmax** | $r^2$ | 0.88 | 0.84 | 0.90 | 0.91 | 0.91 | 0.85 | 0.79 | 0.46 | 0.86 | 0.88 | 0.89 | 0.85 |
| | **RMSE** | 15.77 | 18.01 | 14.50 | 13.08 | 13.54 | 17.50 | 20.63 | 33.06 | 16.63 | 15.43 | 15.04 | 17.50 |
| | **BIAS** | -2.27 | 2.79 | -4.72 | -5.21 | -4.50 | -7.49 | -1.64 | 2.92 | -4.22 | -5.55 | -3.85 | -7.49 |
| **dHS1** | $r^2$ | 0.89 | 0.73 | 0.81 | 0.79 | 0.85 | 0.93 | 0.78 | 0.66 | 0.78 | 0.63 | 0.83 | 0.93 |
| | **RMSE** | 18.64 | 28.64 | 24.04 | 25.56 | 21.49 | 14.84 | 25.71 | 32.12 | 26.17 | 33.66 | 22.96 | 14.84 |
| | **BIAS** | 3.68 | 17.47 | 14.74 | 18.44 | 14.60 | 5.24 | 6.63 | 19.53 | 16.69 | 23.27 | 15.96 | 5.24 |

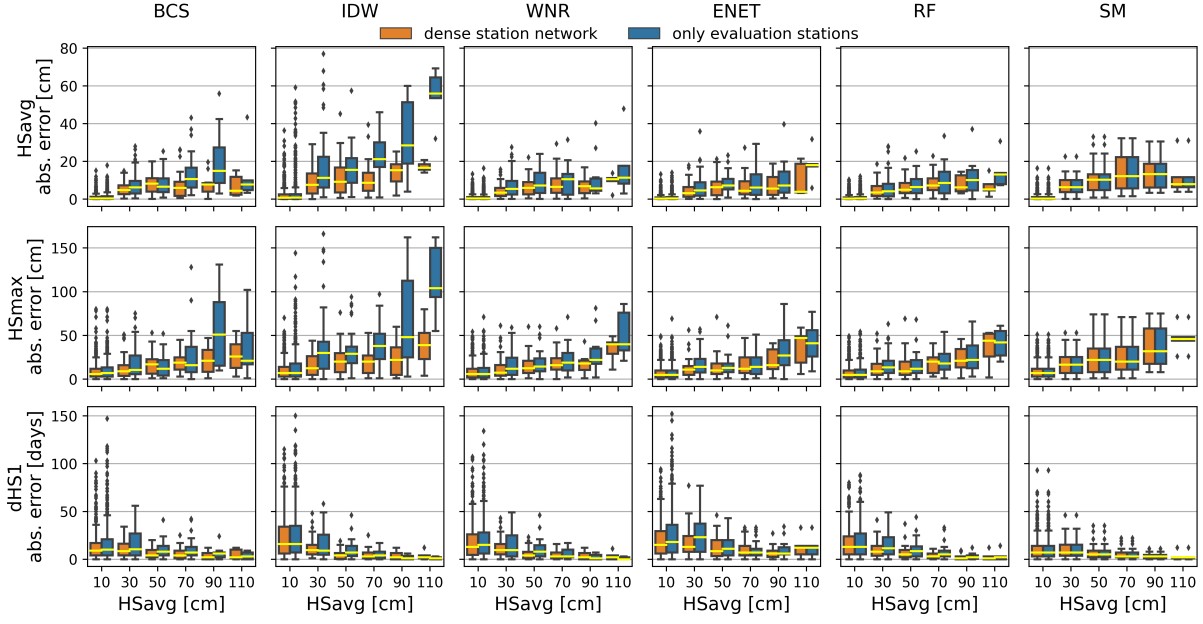

**Figure 5.** Boxplots of absolute errors in average snow depth in a winter (HSavg, top row), maximum snow depth in a winter (HSmax, middle row) and number of snow days with HS≥1 cm (dHS1, bottom row) calculated for 20 cm HSavg bins of the respective gap-winter and the different methods (columns) respectively. Colors of the boxplots denote the different station networks that have been used for reconstruction. Outliers in the boxplots are not shown for better comparison.

## 3.3 Applicability and limitations

Snow depth appears to be a good-natured parameter with respect to reconstructing missing data. All methods except of IDW are able to reconstruct HS with a coefficient of determination above 0.8 regardless of the two station networks used. When deciding what method to choose, it depends on the use case (daily values or derived annual climate indicators) and the setting (station network, surrounding topography, gaps in neighboring stations) in which one wants to reconstruct the data. A qualitative assessment for the suitability of the different methods in different situations and for different applications is given in Table 3.

In a very dense station network such as the one in Switzerland, BCS is able to reproduce annual snow climate indicators HSavg, HSmax and dHS1 with $r^2$ above 0.8 and RMSE below 10 cm for the reconstructed daily HS values. This performance could probably be improved with more advanced bias correction of the neighboring station such as quantile-mapping (Gudmundsson et al., 2012). However, simple approaches such as BCS, IDW and to a smaller extent WNR are sensitive to the density and representativity of the station network. While this is true for every method that uses neighboring stations, more sophisticated methods such as ENET and the nonlinear RF regression are able to almost retain skill also for sparser station networks. Consequently, ENET and RF are besides the SM the most promising candidates in regions with a sparser station network.

Simple spatial averaging with IDW is not able to resemble strong gradients that are present in an alpine topography. We therefore also tested the gradient-plus-inverse-distance-squared (GIDS) method (not shown in results) introduced by Nalder and Wein (1998) which was used in a project of the Austrian Meteorologic service for imputing gaps in HS time-series (Schöner and Koch, 2016). In the sparse network GIDS performed even weaker than IDW, which is in accordance with Price et al. (2000) who observed poor results with GIDS for temperature and precipitation reconstruction in areas with strong topography. Nalder and Wein (1998) compare GIDS to Kriging based methods. We also expect a strong dependence on station network density for Kriging and therefore refrained from including these kind of methods in our method comparison. However, in dense station networks, Kriging can be an alternative approach to our proposed methods for interpolating snow depth data, especially when it comes to spatial continuous reconstructions and not only estimations on a single point.

Buchmann et al. (2021) evaluated the natural variability of annual snow climate indicators by comparing data from parallel station pairs (<3 km distance and <100 m elevation difference). They find RMSE for HSavg within a station pair to be in the same range as RMSE for reproduced HSavg with the ENET, RF, WNR and SM methods. This proves that HSavg can be reproduced reasonably well with these four methods. Even the best performing method in our comparison study cannot reach the quality of a parallel station pair for HSmax and dHS1. RMSE of the RF method is 2 respectively 4 times larger then the median RMSE within a parallel station pair for these two snow climate indicators (Buchmann et al., 2021).

For all methods, highest median absolute errors and BIAS for dHS1 can be observed in winters with low HSavg. These winters are often characterized by an ephemeral snow cover which is building up and vanishing again in the course of the winter. Temperature index models are prone to have problems with these kinds of snow covers which could explain the weaker performance of the SM method in these conditions (Hughes and Robinson, 1993; Gray and Landine, 1988). The positive BIAS of dHS1 for the methods that use several neighboring stations may be explained as following. The probability that at least one

**Table 3.** Suitability of different methods in different situations (dense and sparse station networks, gaps in neighboring stations) and for different applications. Suitability ranges from not recommended or not possible (- -) to very good (++).

|  | BCS | IDW | WNR | ENET | RF | SM |
|---|---|---|---|---|---|---|
| gaps in neighboring station(s) | - - | + | + | - - | + - | ++ |
| sparse station network | - - | - - | - | + | + | ++ |
| dense station network | + | - | + | + | ++ | - |
| daily data | + - | - | + | + | ++ | + - |
| HSavg | ++ | - | + | ++ | ++ | + |
| HSmax | - | - - | + | ++ | ++ | - |
| dHS1 | + | - - | - | - | - | ++ |

of the neighboring stations has snow at a certain day is higher than the probability of snow at the target station. Since most of the methods combine data of the neighboring stations, this will result in statistically more days with snow. When trying to minimize BIAS in dHS1, it is therefore best to rely on only few neighboring stations. Accordingly, BCS yields predictions for dHS1 that have a lower positive BIAS. One possible approach to reproduce dHS1 more accurately than deriving it from reconstructed daily values, could be to model dHS1 directly. This could be realized by fitting a nonlinear statistical model such as random forest to the dHS1 series of the target station with dHS1 series derived from neighboring stations as predictors. However, the reduced number of data points would ideally require a longer training period of simultaneous measurements at target and neighboring stations, respectively. The number of snow covered days can be defined with different thresholds. While a large positive BIAS for the 1 cm threshold (dHS1) can be observed for all methods, this BIAS decreases with increasing thresholds for the snow covered days (see Table A1). For the number of snow days with HS$\geq$10 cm (dHS10) the BIAS is less than 2 days for all methods and decreases further for the number of snow days with HS$\geq$30 cm (dHS30). The coefficient of determination also increases with increasing snow days threshold.

An option to increase the skill of the deterministic methods BSC, IDW and WNR, is to apply stricter constraints to the neighboring stations as done in (Matiu et al., 2021) by introducing a correlation constraint to the neighboring stations (see Section 2.1.4). In the station networks applied in this study, this would lead to a failure in filling data in 15 and 20% of the filled gaps (station-years) for the dense and sparse station network, respectively. These cases occurred mostly for stations at low elevations (AIG, ALT, GVE, SIO, VIS, see Fig.1) with an ephemeral snow cover.

Due to semi-automatic quality control procedures and careful station preselection, our test data set did only contain very few missing HS values for the reference stations. However, this is rather unlikely to be encountered in a real application. Missing values in neighboring stations can be handled differently by different methods. ENET does not allow a single missing value in one of the neighboring stations in the train and gap period. On the other hand, RF and the WNR method are able to deal with missing values in the predictor stations which is a huge asset when it comes to applicability. The effect of missing values in neighboring stations on the performance has not been tested in this study. However, this is an important point to keep in

mind when trying to apply any of the evaluated methods. For RF, it is also possible to add other non snow depth categorical or continuous predictors such as the mean HSavg anomaly of the predictor stations or prevailing large scale atmospheric conditions in the winter of interest. We tested a RF version with an additional categorical predictor calculated from binned quantiles of the mean of all used predictor stations but did not see any improvement over the simpler version using only the season as categorical predictor.

One potential limitation of the SM approach is, that if the snow measurements are interrupted at a certain station, possibly other variables which are needed as input for the snow model could also be missing. However, this is a rather unlikely case to encounter at least in the dataset of Switzerland. Temperature and precipitation traditionally have a higher priority for weather services than the variables associated with snow and therefore in case an issue occurred at a station, the probability of continuation of these two classic meteorologic variables is higher than for any snow variable. After the automation of many weather stations (not for snow) in Switzerland in the 1980s, long gaps in the temperature and precipitation record are even less likely to be encountered. If other variables such as wind and incoming short- and longwave radiation are also available at high temporal resolution for a station, a more sophisticated snow model such SNOWPACK (Bartelt and Lehning, 2002; Lehning et al., 2002) or CROCUS (Brun et al., 1989, 1992) would probably improve the performance of the gap reconstruction. These physics based models cover processes such as erosion by wind and are thought to better represent settling and melting than the very simple approach used in our study. However, the required input data is, if at all, only available in the most recent decades.

A general limitation of our analysis may be the fact that the sparse station network is still dense when compared to station networks present in other regions of the world (Gubler et al., 2017). If the station network is sparser than in our example, the snow model and RF should be favoured over the other approaches as these both methods show the least sensitivity to station network density in our analysis. Especially in data sparse regions, the probability of having temperature and precipitation data available is much higher than for snow depth observations which points towards the use of a snow model for data reconstruction. Alternatively, one could make use of output from reanalysis products such as ERA5-land (Muñoz Sabater et al., 2021). If available, snow depth can be used directly from the reanalysis product or other meteorologic variables from the reanalysis product can be used to model snow depth with a snow model. In either way, some sort of downscaling is necessary since reanalysis products are available in a spatial resolution of about 10 or more kilometers. This can for example be done statistically by using e.g. the Random Forest model described in Sec. 2.1.6 with data from the 9 surrounding grid cells of the target station as predictor variables. This method would be independent of neighboring stations and can be applied worldwide if a global reanalysis product is employed. However, the low spatial resolution of reanalysis products will always limit the application in complex mountainous terrain. Moreover, reanalysis products often suffer from a temperature bias (e.g. Scherrer, 2020), which is crucial with respect to a variable like the highly temperature-sensitive snow cover.

Ultimately, gap filling is often a preceding step when it comes to data homogenization in order to correct time series that show breaks due to station relocations or changes in measurement techniques or instrumentation (Marcolini et al., 2019). These breaks can be accompanied by a period of missing data. Reconstruction methods that employ training methods before and after a data gap could level out breaks and potentially complicate their detection and correction. Therefore, it might be advisable

to only use a training period from either before or after the data gap. Caution is also necessary when trying to e.g. do break detection on reconstructed annual dHS1 series due to the positive biases introduced by most of the methods.

## 4 Conclusions

We compared different methods for reconstructing long gaps in daily manual HS data records as well as their ability to reconstruct the annual snow climate indicators HSavg, HSmax and dHS1. The ENET, RF, WNR and BCS method are able to

reproduce daily HS values with coefficient of determination above 0.9 in the dense and above 0.8 in the sparse station network, respectively. Median RMSEs of the filled gaps are below 4 cm for all methods. The SM which does not need data from neighboring stations reveals only slightly lower coefficient of determination (0.86) for daily HS values. The two annual climate indicators HSavg and HSmax, in contrast to dHS1, can be well reproduced by BCS, ENET, RF, SM and WNR. All methods except for SM and BCS overestimate the dHS1 with BIAS of 15 to 23 days. In a sparse station network a simple snow model

is best suited to resemble dHS1 most accurately with $r^2$ of 0.93.

The reconstruction errors of HSavg are within the natural variability of a parallel station pair. Snow depth seems to be a relatively good-natured parameter when it comes to gap filling of data with neighboring stations. However, when station networks get sparse, temperature index snow models serve as a suitable alternative to classic inter-station gap filling approaches.

Since most of the methods perform reasonably well, the choice of which method to use depends on the specific use case and

setting. If a serially complete, highly correlated station is available, bias corrected data from this station is easy to calculate and, in many instances, sufficient enough to be used in a climatological use case. If no meteorological data is available at the target station and if neighboring stations regularly contain missing data as well, WNR is a suitable deterministic approach to reconstruct data from neighboring stations. Missing data in neighboring stations can also be handled by RF. If the station network is sparser than in our study and if neighboring stations are further away and weakly correlated, the snow model, ENET

and RF should be favoured over the other approaches as these three methods show the least sensitivity to station network density in our analysis. If the focus of the analysis is set on dHS1, a simple snow model is best suited to reconstruct a complete missing winter. If no meteorological data is available, BCS should be the fallback solution for dHS1 in case a suitable reference station is available.

## Appendix A: Additional figures and tables

*Code and data availability.* Python code to perform the analysis and to use the methods on other data is available from Aschauer (2021). Due to data policies, input data to reproduce the analyses is available upon request from the authors.

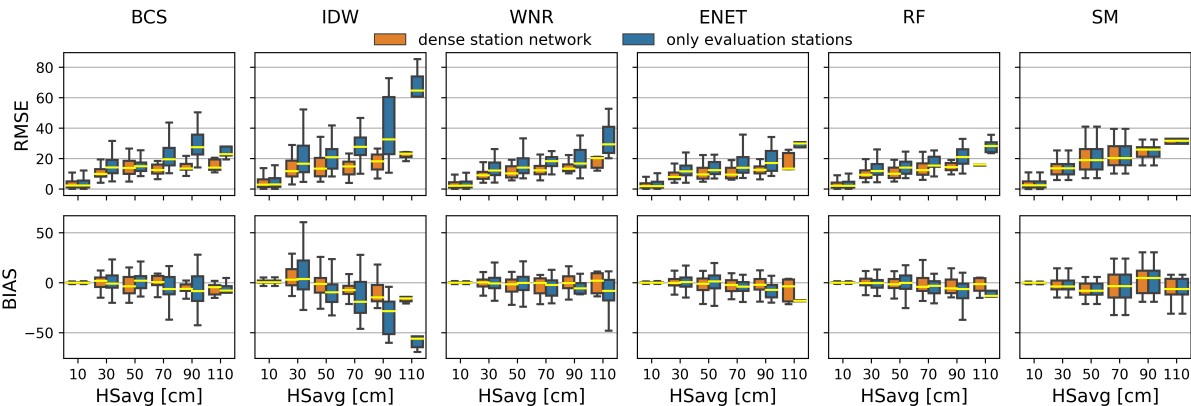

**Figure A1.** Boxplots of RMSE (top row) and BIAS (bottom row) calculated for 20 cm HSavg bins of the respective gap-winter and the different methods (columns) respectively. Colors of the boxplots denote the different station networks that have been used for reconstruction. Outliers in the boxplots are not shown for better comparison. The maximum number of predictors for the different methods is set as defined in Tab. 1.

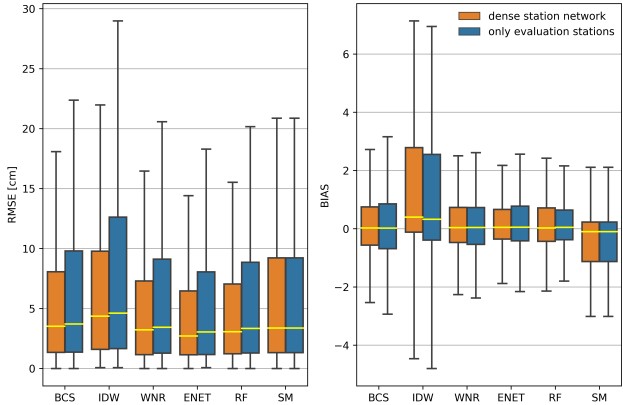

**Figure A2.** Boxplots for root-mean-squared error (RMSE) and BIAS of the daily values for the different methods and station densities. The maximum number of predictors for the different methods is set as defined in Tab. 1. The station network density is irrelevant for the snow model as no data from neighboring stations is used. For better comparison, outliers are not shown in the boxplots.

*Author contributions.* JA and CM designed the study. JA performed the analysis and drafted the manuscript. Both authors discussed the results and commented on the manuscript.

*Competing interests.* The authors declare that they have no conflict of interest.

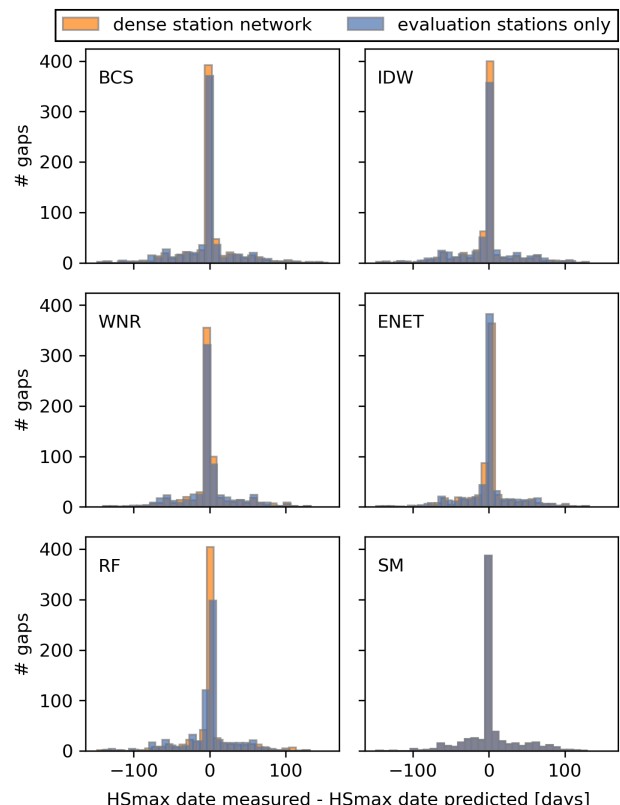

**Figure A3.** Histograms showing the difference in days between the measured date of HSmax and the date of HSmax in the reconstructed winters in the dense and sparse station networks. In case the same HSmax is recorded on more than one day, the date of the first occurrence is taken.

*Acknowledgements.*  We want to thank MeteoSwiss for providing data of their meteorological stations as well as Tobias Jonas for input on the density model. Thank you to Moritz Buchmann for valuable discussions and comments on the manuscript. We want to thank J. Ignacio López-Moreno, Michael Matiu and two anonymous referees, whose valuable comments helped to improve the manuscript.

**Table A1.** BIAS, RMSE and coefficient of determination ($r^2$) for the 1 cm (dHS1), 2 cm (dHS2), 5 cm (dHS5), 10 cm (dHS10) an 30 cm (dHS30) thresholds for the number of snow days reconstructed with the different methods in the dense and sparse station networks.

| | | dense station network | | | | | | evaluation stations only | | | | | |
|---|---|---|---|---|---|---|---|---|---|---|---|---|---|
| | | BCS | IDW | WNR | ENET | RF | SM | BCS | IDW | WNR | ENET | RF | SM |
| **dHS1** | $r^2$ | 0.89 | 0.73 | 0.81 | 0.79 | 0.85 | 0.93 | 0.78 | 0.66 | 0.78 | 0.63 | 0.83 | 0.93 |
| | **RMSE** | 18.64 | 28.64 | 24.04 | 25.56 | 21.49 | 14.84 | 25.71 | 32.12 | 26.17 | 33.66 | 22.96 | 14.84 |
| | **BIAS** | 3.68 | 17.47 | 14.74 | 18.44 | 14.60 | 5.24 | 6.63 | 19.53 | 16.69 | 23.27 | 15.96 | 5.24 |
| **dHS2** | $r^2$ | 0.90 | 0.78 | 0.88 | 0.89 | 0.92 | 0.93 | 0.83 | 0.70 | 0.88 | 0.78 | 0.90 | 0.93 |
| | **RMSE** | 17.30 | 26.16 | 19.22 | 18.86 | 15.60 | 14.94 | 23.43 | 30.62 | 19.76 | 26.12 | 17.44 | 14.94 |
| | **BIAS** | 3.19 | 14.12 | 9.03 | 10.57 | 8.16 | 3.17 | 5.21 | 16.31 | 9.55 | 14.65 | 9.22 | 3.17 |
| **dHS5** | $r^2$ | 0.94 | 0.83 | 0.93 | 0.93 | 0.95 | 0.94 | 0.88 | 0.73 | 0.92 | 0.88 | 0.93 | 0.94 |
| | **RMSE** | 14.31 | 23.33 | 14.96 | 14.58 | 12.06 | 14.14 | 19.37 | 29.25 | 16.22 | 19.90 | 14.74 | 14.14 |
| | **BIAS** | 2.05 | 10.67 | 3.77 | 4.50 | 3.25 | 0.84 | 3.23 | 12.80 | 3.84 | 7.53 | 4.09 | 0.84 |
| **dHS10** | $r^2$ | 0.95 | 0.86 | 0.95 | 0.96 | 0.96 | 0.94 | 0.92 | 0.73 | 0.93 | 0.94 | 0.95 | 0.94 |
| | **RMSE** | 11.99 | 21.27 | 12.64 | 11.26 | 10.57 | 14.24 | 16.20 | 28.92 | 14.83 | 13.61 | 12.28 | 14.24 |
| | **BIAS** | 0.72 | 8.21 | 0.81 | 1.48 | 1.07 | -0.01 | 0.46 | 9.88 | 1.51 | 2.19 | 1.68 | -0.01 |
| **dHS30** | $r^2$ | 0.93 | 0.85 | 0.95 | 0.95 | 0.95 | 0.88 | 0.91 | 0.65 | 0.92 | 0.93 | 0.94 | 0.88 |
| | **RMSE** | 12.90 | 18.79 | 11.29 | 10.51 | 11.01 | 16.55 | 14.69 | 28.55 | 13.62 | 12.53 | 12.14 | 16.55 |
| | **BIAS** | -0.74 | 4.61 | -1.34 | -0.54 | -0.37 | -2.41 | -0.99 | 5.00 | -1.24 | -0.96 | -0.57 | -2.41 |

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
