# Peer review of "Evaluating methods for reconstructing large gaps in historic snow depth time series"

_Geoscientific Instrumentation, Methods and Data Systems, 2021_

## Author Comment (AC1)

**Author response to Comment CC1 on gi-2021-16**

Thank you Michael for your time to give the manuscript a read and for your valuable feedback suggestions. Below, we respond to each suggestion and comment one by one. The reviewer comments are highlighted in blue while our responses are kept in black.

I've been following this work since over 1 year and am happy it's being published. Congratulation to the authors for this evaluation, which is immensely helpful for future climatological assessments of station observations of snow depth. Also, they are to be thanked for making the code publicly accessible.

I suggest using consistent naming of the Matiu/WNR method in figures, tables, and text. Personally, I prefer WNR, since it's not "my" method…

[Answer]: We named the WNR method consistently throughout the manuscript.

The comparison of more traditional approaches to ML tools is very useful.

The only thing that I found odd is the large bias and errors in dHS1. A higher one for HSmax is to be expected, because one value (maximum) can behave very sensitive. But if daily HS is reconstructed well, as well as HSavg, why not dHS1? However, I reproduced the results of dHS1 based on Matiu et al. 2021 and found basically the same order of bias in dHS1 for the WNR method. Even though, it appears to be highest in middle elevations (1000-1500m) and lower otherwise, and the bias decreases with a higher SCD threshold, e.g it's halved for 2cm, and becomes negligible (almost 1/6) for 5cm. There seems to be a minimal positive bias for low HS in daily reconstructions, negligible for HSavg, but enough to introduce errors in dHS1.

[Answer]: We thank for this input and just want to remind that HSmax, despite being one single value, usually is the result of snowfall accumulation over several days or months. Regarding dHS1, we calculated the number of snow days for thresholds of 1, 2, 5, 10 and 30 cm (dHS1, dHS2, dHS5, dHS10, dHS30). As you write above, the BIAS decreases with increasing threshold for snow covered days. We added some bits to the discussion and included Table A1 in the appendix.

Initially, we also had the suspicion that the positive BIAS for dHS1 could arise from the fact that our methods are rounding predictions to the nearest integer and that accordingly values between 0.5 and 1 cm are still contributing to dHS1. However, after having a look at dHS2 which is still positively biased for all methods except of BCS, we believe it is unlikely that rounding to integers is the source of the large positive biases and did not do further investigations towards that direction.

Maybe the authors could provide a table in the applicability and limitations section to summarize the reconstruction methods evaluated? Showing e.g. "Best/Good", "OKish", and "Not recommended" methods, depending on parameter (daily, seasonal, …) and network density? I know, this might involve some arbitrary choices, but could be useful nonetheless. And, this could highlight the dHS issue and warn against using IDW for snow - in addition to what the authors already write in the text…

[Answer]: We added an overview table (Table 3) to the discussion which summarizes the suitability of different methods in different situations (dense and sparse station networks, gaps in neighboring stations) and for different applications.

---

## Author Comment (AC2)

**Author response to Comment RC1 on gi-2021-16**

We thank the reviewer for her/his time to assess the manuscript and for the valuable feedback suggestions. Below, we respond to each suggestion and comment one by one. The reviewer comments are highlighted in blue while our responses are kept in black.

The manuscript "Evaluating methods for reconstructing large gaps in historic snow depth time series" by Aschauer and Marty is an interesting technical study focusing on interpolation of snow depth time series. The topic is relevant for the community focusing on the snow modeling and time series analysis. The paper summarizes in an effective way concepts which are known by the community, by applying six different interpolation methods to a snow dataset from Switzerland. I think the manuscript represents a good contribution to the current scientific discussion on the topic. I have a few suggestions that the authors could consider in the revised version of the manuscript:

- Lines 43-45: I only partially agree with this statement. The fact that temperature index models require only precipitation and temperature data is the only reason why someone should apply them. In fact, they provide only a very rough physical representation of snow processes. Moreover, it seems that the authors do not consider in the modeling approach processes such as snow redistribution driven by wind or gravitation. These processes are generally not dominant if the choice of the location of the monitoring station is performed properly, but the authors may comment on it.

[Answer]: We clarified the sentence and added two additional sentences in which we take up the points from your comment and highlight the simplistic nature of temperature index models and the associated problems.

- Line 70: I think time series available for this dataset are generally longer than the period selected by the authors (1999-2020). Could they provide shortly a justification for this choice?

[Answer]: Yes, many stations from our station set cover longer time ranges than 1999-2020. We wanted to have as many gap free station records in our set of evaluation and predictor stations as possible and therefore decided to take the last 21 years because the most recent period has the highest station density in the Swiss station network. Additionally, the 21 year time period was chosen because we wanted to have a  long enough data set on the one hand (containing a few well known snow  abundant and snow scarce years) and a common (realistic) length of available snow depth time series for the training period on the other hand.

We think that 21 years per station is enough for the cross validation. If the period would be longer, the "leave one winter out" approach would probably become problematic due to potential shifts in snow climatological dependencies since many of our models assume temporal stationarity. We added two sentences to the manuscript to clarify our choice of the selected period for our station dataset.

- Line 87: should "station or both" be "station of both"?

[Answer]: Yes, has been corrected to "station of both".

- Interpolation methods: I fully understand that it is not possible to test all available interpolation methods, but I was surprised that a standard interpolation method like Kriging with external drift

or let's say Kriging based methods were not considered. Could the author please motivate this choice?

[Answer]: As stated in the discussion [line 265 ff] we also tested the GIDS method during early phases of the manuscript preparation. In the reference paper of GIDS [https://doi.org/10.1016/S0168-1923(98)00102-6], the authors also compare this technique to Kriging based methods. The findings there are pointing towards the direction that Kriging is not superior to GIDS and also has problems in sparse station networks.

However, when trying to do spatially continuous interpolation of snow depth data and not only for a single target (i.e. a single station), Kriging might be a suitable approach. We added a part to the discussion section where we explain our reasoning on why we did not include Kriging based methods in our set of compared methods.

- Line 178: Is a threshold of 1 cm really relevant for tourism?

[Answer]: In fact, for winter tourism higher HS thresholds such as 30 cm are more often used [e.g. https://doi.org/10.5194/tc-13-1325-2019]. However, as the winter tourist expects a nice wintery scenery, a threshold of 1 cm and the question whether a snow cover is present has also some importance.

- The Matiu/WNR method should be consistently named in the manuscript

[Answer]: We named the WNR method consistently throughout the manuscript

- Line 230 should "derived form" be "derived from"?

[Answer]: Yes, has been corrected accordingly.

- Line 237 I think that beside the value of HSmax it would be interesting to compare when HSmax occurs using the different methods.

[Answer]: We calculated the difference in days of the date of HSmax measured - date of HSmax reconstructed. We added Figure A3 with the distributions of these differences to the appendix and inserted a sentence in the results section.

- Figure 4: it is very hard to read the values of r2, RMS and BIAS, please move them to a Table

[Answer]: We removed the numbers from the figure and moved them to the newly added Table 2.

- Although I acknowledge the difficulty of summarizing the results of this work in a graphical form, I find the quality of Figures 4, 5 and A1 low. The black lines in the blue box plots are hardly readable and in general the size of the subplots it is too small allow the reader a quantitative interpretation of the results.

[Answer]: We changed the color of the median lines in the boxplots in order to make them appear more visible. Additionally, we made the subplots a little larger by getting rid of in-between labels.

- Line 298: It would be nice to have a ranking of the methods to be applied in these circumstance.

[Answer]: We added an overview table (Table 3) to the discussion which summarizes the suitability of different methods in different situations (dense and sparse station networks, gaps in neighboring stations) and for different applications.

- The authors correctly point out that data interpolation is an important step to have longer time series for climatological analysis. However, after data interpolation the next step would be to homogenize stations. Homogenization of snow depth time series is an actual research topic (e.g., Marcolini et al., 2019) and it would be nice to see in the discussion part some comments given by the authors about the effect of the different interpolation method on the quality of the resulting time series. I do not expect a quantitative assessment, since it would probably result in another paper due to the required amount of work, however some qualitative insights would be interesting to stimulate further research in this direction.

[Answer]: You are correct; often gap filling is a precursory step of homogenizing a climatological measurement series. We added a paragraph in which we qualitatively assess the effect of filling gaps on homogenization.

Marcolini, G., Koch, R., Chimani, B., Schöner, W., Bellin, A., Disse, M., & Chiogna, G. (2019). Evaluation of homogenization methods for seasonal snow depth data in the Austrian Alps, 1930–2010. *International Journal of Climatology*, *39*(11), 4514-4530.

---

## Author Comment (AC3)

**Author response to Comment RC2 on gi-2021-16**

Thank you Mr. Lopez-Moreno for your time to assess the manuscript and for your valuable feedback suggestions. Below, we respond to each suggestion and comment one by one. The reviewer comments are highlighted in blue while our responses are kept in black.

The manuscript presents the comparison of different methods to fill gaps in snow series. This is a task that has generated many doubts to snow reseachers and this paper provides very useful information for readers. The paper has a clear structure, is well written and conclusions are sound and clear. Therefore, I recommend the publication of the article, with just a few comments that authors may consider to prepare a revised version of the manuscript.

1- in my opinion, it would be interesting to present some analysis to show how differenet methods are suitable to fill gaps of different length, as probably there will be important differences among accuracy scores and methods.

[Answer]: We originally planned to include different gap lengths in our analysis but decided to use only a unique gap length (one winter) for our method comparison. Additional gap lengths require decisions regarding the training period and how the gaps are created. Furthermore, the amount of results blows up and it will become more difficult to compare methods easily. This is why we decided to keep the study simple and only consider long gaps of a whole winter of missing data.

2- As you can include other categorical variables in the Random Forest, authors can test or at least discuss other possible predictors that might refine the results. In example clasiffy if gaps occur in low/average/high snow years; or it existed different dominant weather types or atmospheric patterns in a given year when gaps must be filled.

[Answer]: During early phases of manuscript preparation, we tested different versions of Random Forest interpolation where we also included other categorical variables such as binned quantiles of the mean of all used predictor stations. These versions did not add improvement to the simpler RF version we present in the paper. However, it is worth noting that this kind of predictors are also possible to be used in Random Forest interpolation and might be useful in certain circumstances. We added a corresponding sentence to the discussion.

3- Authors may discuss to which extent the use of more physically based (when possible) may improve the error estimators compared to the degree day model. Researchers from CEN uses adjusted crocus/safran simulation to fill gaps in snow series (see https://doi.org/10.1002/joc.6571 as example). In a similar way bias corredted ERA-land series could be used for some areas, or used as "virtual" best correlated stations.

[Answer]: We added a paragraph to the discussion where we cover the potentially beneficial use of more physics-based snow models such as SNOWPACK or CROCUS if the necessary data is available.

Additionally, we added a paragraph in which we discuss the possibility of using reanalysis data instead of neighboring station data for the spatial interpolation methods or as input for the snow model.

Despite of the fact that we believe it is out of scope of the paper to assess estimations of snow depth data from reanalysis products, we did a quick assessment of the potential using ERA5-land reanalysis data. We tested three different schemes for gap reconstruction:

1. ERA5-land snow depth data without any bias correction from the closest grid point (ERA5nobc)
2. ERA5-land snow depth data from the closest grid point with the same mean ratio bias correction applied to the BCS method (ERA5mrbc)
3. ERA5-land snow depth data from the 9 surrounding grid points as input to the RF method (ERA5rf)

We applied the same leave-one-winter-out cross validation at the evaluation stations as for the other methods. The scores for HSavg, HSmax and dHS1 are listed in the following table:

|  |  | ERA5nobc | ERA5mrbc | ERA5rf |
|---|---|---|---|---|
| HSavg | r2 | -10.51 | 0.84 | 0.86 |
|  | RMSE | 73.69 | 8.68 | 8.14 |
|  | BIAS | 53.13 | -0.02 | 0.25 |
| HSmax | r2 | -3.99 | 0.5 | 0.78 |
|  | RMSE | 100.16 | 31.69 | 21.08 |
|  | BIAS | 72.21 | -21.62 | -6.84 |
| dHS1 | r2 | -1.74 | -0.6 | 0.52 |
|  | RMSE | 91.51 | 69.74 | 38.05 |
|  | BIAS | 78.51 | 56.46 | 27.82 |

These single grid point approaches perform clearly inferior to all tested methods in the paper. The Random Forest downscaling approach (ERA5rf) can compete with IDW for HSavg and HSmax but is not able to reach the performance of the other methods. dHS1 is more biased than any other method with BIAS of +27.8 days. For the rare case, that meteorological data is also missing when snow data is missing, it would be interesting to first downscale temperature and precipitation and then use that as input for a snow model. However, as stated in the discussion, the probability that there are gaps in temperature and precipitation is lower than for snow. In case a local measurement is available as input for a snow model, we believe this will always be superior to the reanalysis driven approach.

Moreover, reanalysis products often suffer of an elevation dependent precipitation or temperature bias, which is crucial in regard to a highly temperature sensitive variable as snow cover. Additionally, higher resolution reanalysis products like ERA5 are not available for the historic gaps in the first half of the 20[th] century. Nevertheless, it would be interesting to assess the potential of different reanalysis products for snow depth reconstruction in a follow up study.

Looking forward to see your revised manuscript,

Ignacio López-Moreno

---

## Author Comment (AC4)

**Author response to Comment RC3 on gi-2021-16**

We thank the reviewer for her/his time to assess the manuscript and for the valuable feedback suggestions. Below, we respond to each suggestion and comment one by one. The reviewer comments are highlighted in blue while our responses are kept in black.

In their manuscript "Evaluating methods for reconstructing large gaps in historic snow depth time series", the authors compare different methods for filling large gaps in measured snow depth time series in Switzerland.

The manuscript is very well written and of high technical and scientific quality. The comparison of the presented methods is in general of interest for the respective snow hydrological community, however, in my opinion only for a very small number of real use cases. The authors show that already a very simple snow model approach using measured temperature and precipitation as input can yield more or less the same results. There are much more temperature and precipitation measurements available than snow depth observations, especially in data sparse region. For that reason, I don't see very much applicability of the results.

[Answer]: We only partly agree. Yes, we cover quite a specific use case with our study, which is the reconstruction of long-term gaps in historic snow depth time series. We indeed believe that for this and similar use cases our method comparison is of large value to the scientific community. For example, the homogenization community is often confronted with the problem of such gaps in the step of the break-detection (see the newly added paragraph at the end of the discussion section). Solely the conclusion that a simple temperature-index snow model is able to represent a decent amount of variability when it comes to reproduce snow climate indicators such as HSavg, HSmax or dHS1 is in our opinion of value. Moreover, there is an increasing number of cases in high alpine environments, where either precipitation is not measured or the measured precipitation amount is strongly limited due to under-catch, but on the other hand a large number of neighboring snow stations is available due to specific needs like avalanche warning.

As the authors state, HS is a good-natured variable for gap-filling. This holds true for measurements in terrain where the presented stations are usually located and for continuous snow coverage and typical seasonal, continuous accumulation and ablation dynamics. Therefore, it is quite obvious that good results can be obtained using statistical interpolation methods (more or less regardless of type) using neighboring stations of similar elevation. Much more interesting would be an extension of the analysis to terrain characteristics (lateral snow redistribution, steep terrain, slope, aspect, i.e. small scale heterogeneity in mountainous terrain). This could be tackled by connecting the presented methods to stations clustered not only by elevation and distance, but also slope, aspect, etc. However, I see that this is probably not possible due to the stations located at "representative", flat, unobstructed terrain locations.

[Answer]: This is a very good point because with the extension of the methods to consider terrain characteristics, it would be also possible to interpolate snow to areas where we do not have station information. Unluckily, as you already write, we lack the necessary data to train any method of that kind since the stations are located at sites that do not differ too much from each other regarding slope or aspect. However, we ultimately are interested in getting continuous snow depth time series at a station location. Therefore, despite being interesting, these questions should maybe rather addressed in another study and are out of scope of our study.

Regarding the results of dHS1, it would be interesting to see the same analysis for dHS10 or dHS5, i.e. a threshold for a snow day of 10 or 5 cm snow depth, as 1 cm is within a range of errors/uncertainties of all measuring and modeling methods. Probably the results will be much clearer using a slightly higher threshold.

[Answer]: We calculated the number of snow days for thresholds of 1, 2, 5, 10 and 30 cm (dHS1, dHS2, dHS5, dHS10, dHS30). We added a paragraph to the discussion regarding the effect of different snow cover day thresholds and included a table (Table A1) in the appendix.

As pointed out by reviewer 2, the study would highly benefit from an additional comparison to derivates or direct model values from, e.g. reanalysis products, which are readily available globally.

[Answer]: Please see our answer to RC2 regarding the use of reanalysis products. An assessment of downscaling methods for snow depth or temperature and precipitation from different reanalysis products would get to broad and would be out of scope of this paper. Nevertheless, we added a few sentences to the discussion section where we discuss the potential of using reanalysis data but refrain from including it to the results.

I support the idea raised in the other comments of including a table with particular strengths and weaknesses of the methods depending on the application and data availability.

[Answer]: We added an overview table (Table 3) to the discussion which summarizes the suitability of different methods in different situations (dense and sparse station networks, gaps in neighboring stations) and for different applications.

Apart from the – in my opinion – rather low applicability of the presented results in other scientific use cases, the article presents a technically well performed study. The findings and conclusion are presented in a very clear and concise way.